# OpenReview forum: "CleanGen: Mitigating Backdoor Attacks for Generation Tasks in Large Language Models"
_ICLR.cc/2025/Workshop/BuildingTrust — BuildingTrust_

### Official Review · Reviewer_CABc · 2025-02-21
**Good paper, with a few objections**

**Rating:** 8
**Confidence:** 3

**Review:**

Pros:
- This is a good paper.  The defense seems highly practical and effective.
- The paper is well written, there are extensive evaluations, and the results look strong
- The graphics look nice.

Questions:
- How does this defense work with transfer attacks?  In this paper, it seems like you optimize the attacks against the target model.  Clever attackers often optimize attacks against a large number of models to encourage transfer.  If both the target and the defender model are susceptible to the attack, then your defense will likely fail.
- The inference cost of your defense is clearly high, since you need to evaluate using a secondary language model.  Is there any way that you can be selective about soliciting feedback from the defender model?  Does the defender model need to have any properties to guarantee robustness?

---

### Official Review · Reviewer_QqsG · 2025-02-24
**Interesting backdoor detection method supported with clear insights and comprehensive results**

**Rating:** 7
**Confidence:** 4

**Review:**

The paper develops a method for detecting and mitigating backdoor attacks for generation tasks. In particular, the method leverages a reference model assumed to be separately fine-tuned on clean datasets (or poisoned using a different method) and builds upon the insight that targeted response tokens have significantly higher sampling probabilities compared with the target backdoored model (compared with the reference model). Comprehensive experiments support the superiority (effectiveness, helpfulness, efficiency) of the proposed method.
The paper is well-written and well-structured. The key messages and the positioning of this paper within the existing literature can be easily understood by readers. The designed method is sensible, and the insights are clearly explained. The experiments are comprehensive, and the results suggest the proposed method's strong performance.

Nevertheless, I have a main question about the setup in which a "clean" model can be properly obtained (in addition to the backdoored target model). Are you using the same distribution of (benign) data samples to create the target backdoored model and to fine-tune the reference model? If this is the case, I'm not sure how realistic this setup is. In my opinion, the most realistic scenario is that either the model trainer only has access to a poisoned fine-tuning dataset, regardless of creating the target model or the reference model, or the model trainer resorts to some alternative easy-to-acquire "clean" dataset but likely comes from a different distribution. Providing and motivating the precise set of assumptions on the training setup of the reference model is recommended. In addition, the paper can discuss the potential adaptive backdoor strategies that may reduce the effectiveness of the proposed detection/defense methods.

---

### Official Review · Reviewer_ydbd · 2025-03-02
**While the paper does propose an effective backdoor defense of backdoor generation addressing some of the concerns can enhance the paper.**

**Rating:** 6
**Confidence:** 4

**Review:**

Overview:

This paper presents a novel defense against backdoor attacks in generation tasks within NLP. The core idea is to use a reference model to compute a ratio between the probabilities assigned by the poisoned model and the reference model. Based on this ratio, a decision is made on whether to permit the token or not, thereby performing rejection sampling with the reference model.

Positives
1. In terms of complexity, this is a viable solution. For instance, if paired with smaller and larger models, it can even outperform standard autoregressive generation, as the defense functions as a slight variation of speculative decoding.
2. The experiments provide sufficient evaluation of the proposed attacks.
3. The paper is well-written and easy to read.

Concerns
1. Why is speculative decoding not viable as a defense given the presence of a reference model? In speculative decoding, when the reference model's probability is lower than the backdoor model's probability, the token is permitted with a probability of (reference/backdoor), which can be very small if the inverse of these probabilities (ratio in this paper) can be very large. While it is possible that, if the ratio is not extreme, some bad tokens may still be permitted, this brings me to the second point—generation may end up being upper-bounded by the utility of the reference model. Providing insights into this would be helpful.

2. Another concern is the lack of analysis on whether the proposed generation approach is upper-bounded by the reference model's performance. Future versions of the paper could benefit from additional results addressing this. While I understand that if only backdoor-generated tokens show a massive difference in the ratio, most of the generation will still be guided by the backdoor model, thus preserving clean generation performance, a study into the distribution of the ratio between bad tokens and general tokens would be insightful.

3. The method essentially relies on having a strong reference model, as it hinges on comparing the likelihood of a token in the reference model versus the backdoor model. While it is reasonable to assume that the likelihood of bad generations will differ across two different models, the possibility of the same backdoor attack affecting both models cannot be ignored. This is particularly relevant because backdoor attacks are often implemented through fine-tuning, and if the fine-tuning dataset is public, it could potentially be used to fine-tune both models. If a non-fine-tuned model is chosen as the reference model, it may negatively impact generation quality, as the response will be heavily influenced by the reference model's behavior.

---

### Decision · Program_Chairs · 2025-03-04

Accept